# Hybrid Apparent Diffusion Coefficient (HADC) Map

**Author(s) names withheld**                                                        EMAIL(S) WITHHELD

## Abstract

Multiparametric MRI (mpMRI) is an established framework for prostate cancer assessment which includes T2-weighted magnetic resonance (T2w-MR) and diffusion weighted (DW) sequences. Low quality of Apparent Diffusion Coefficient (ADC) maps from the diffusion sequence can hinder such clinical assessment. Herein, we propose to generate Hybrid ADC (HADC) maps from high-quality T2w-MRI using "lesion-aware cycle-consistent generative adversarial network (LA-CGAN)". Our produced HADC maps contain anatomical information from T2w-MR, and high intra-prostatic contrast of cancerous vs normal tissue, similar to the acquired ADC. Initial results have satisfied the expert radiologist in producing HADC. This work can considerably improve the quality of mpMRI combined assessment for prostate cancer detection.

**Keywords:** Prostate cancer detection, Hybrid ADC map, lesion-aware Cycle-GAN

## 1. Introduction

**Background**   An estimated one in nine American men will have prostate cancer during his lifetime, making it the second leading cause of cancer-related deaths among American men (Simon, 2019). While challenging, early diagnosis of prostate cancer can prevent deadly cancer metastasis. Recently, multiparametric magnetic resonance imaging (mpMRI) of the prostate have become well-established for prostate cancer diagnosis (Harmon et al., 2019). An mpMRI is composed of: dynamic contrast enhanced (DCE), diffusion weighted image (DWI) sequences, along the conventional T2w-MR sequences (Maurer and Heverhagen, 2017). Using mpMRI has considerably improved noninvasive visualization of suspicious lesions for more accurate detection, localization and assessment of malignant foci in the prostate (Maurer and Heverhagen, 2017). However, DWI has proved to be the most effective in differentiating between cancerous and non-cancerous tissue. More specifically, literature suggests utilizing Apparent Diffusion Coefficient "(ADC)" map as a quantitative parameter of DWI due to its hypo-intense imaging signal within densely packed tumor cells (Harmon et al., 2019).

**Motivation**   While T2w-MRI contains detailed anatomic clues about cancerous lesions, it can be difficult for radiologist to visually risk-assess lesions with this sequence alone. Opposingly, even without any anatomical detail, ADC map has an informative lesion expression by significantly lower mean values in cancerous tissue compared to benign areas. As a result, most studies use both the ADC map along with its respective T2w to locate prostate cancer lesions (Harmon et al., 2019). Ideally, looking at two sequences simultaneously, radiologist is expected to cognitively link the functional ADC map to the reference anatomy in T2w-MR,

in order to draw a conclusion regarding cancer risk of lesions in prostate. Unfortunately, ADC maps are prone to distortion, blurriness, and low signal to noise ratios, secondary to anatomical non-synchronization and/or motion artifacts which make them non-diagnostic and difficult to interpret. That is why such an assessment requires a considerable level of expertise from radiologists which leads to inter- and intra-reader variability. It is also a time-consuming process to obtain ADC images in addition to T2w-MRI. Luckily, there is a great body of evidence that confirms the information content of T2w images overlaps that of ADC maps within lesions (Gibbs et al., 2009; Langer et al., 2010). While impossible to extract the ADC map from T2w-MR images with naked eyes, DW information is contained within T2w images which provides the potential to extract this information using what we call a "Hybrid ADc (HADC)". We use "Hybrid" to emphasize that our generated HADC maps inherit anatomical information from T2w-MRI while representing informative lesion contrast as in acquired ADC.

**What we propose** To the best of our knowledge, we are the first to generate HADC map from T2w-MR sequence using our proposed LA-CGAN (Zhu et al., 2017). By introducing HADC we aim to reach two goals: 1) to produce distortion free ADC maps which preserve the anatomic structure of the prostate, 2) to provide an assisting tool for improved prostate lesion assessment, with a potential to substitute acquired ADC maps to have quicker, less costly exams.

## 2. Related work

The idea of image to image translation has brought lots of potential into medical image processing for radiology (Kazeminia et al., 2018; Nie et al., 2017; Wolterink et al., 2017a; Jiang et al., 2018; Wolterink et al., 2017b). Starting from non-parametric models in 2000 and parametric translation using CNNs in 2017 trained by paired input-output images, these methods have been evolved into unpaired Cycle-GAN (Isola et al., 2017; Zhu et al., 2017). Using CycleGAN (CGAN) has overcome the obligation to have anatomically registered pair of images for training (Zhu et al., 2017). Herein, we proposed unpaired image-to-image translation to map anatomical into functional images using our designed architecture LA-CGAN.

## 3. Proposed Methodology

Building upon unpaired CGAN (Zhu et al., 2017), we designed LA-CGAN for dual-domain image translation among images from $A$-space (T2w-MR) and $B$-space (ADC map). Our proposed framework is composed of two generators: $\mathbf{G}_A$ generating $I_{BA}$ from $I_B$, $\mathbf{G}_B$ generating $I_{AB}$ from $I_A$, a U-net segmentation network to segment lesions in $I_{AB}$ learning from labels in $I_A$, and two discriminators: $\mathbf{D}_A$ recognizing $I_A$ from $I_{BA}$, $\mathbf{D}_B$ recognizing $I_B$ from $I_{AB}$ (Fig. 1).

In our design, we employed cycle consistency to ensure domain adaption among T2w-MR and ADC images. Additionally, we restrict the network to preserve the lesion-specific features going from T2w to ADC, utilizing a U-net that learns to segment lesions in the produced HADC map ($I_{AB}$) based on labels in its reference T2w-MR image ($I_A$). Enforcing this segmentation network into our generator structure, our loss function is given as in

Eq. 1, which guarantees structural registration while transforming the lesion-emphasized texture between HADC and the acquired T2w-MR. To facilitate the learning process of our LA-CGAN we applied a prepossessing strategy which results in meaningful intensity values among different MR images: we first used N-4 Bias Correction algorithm to remove low frequency non-uniformity or "bias field" present in MR scans. We then performed a patient level standardization of intensity distribution, where intensity values in each image were transformed such that the resulting histogram holds in harmony with a predefined standard distribution (Tustison et al., 2010), (Nyúl and Udupa, 1999).

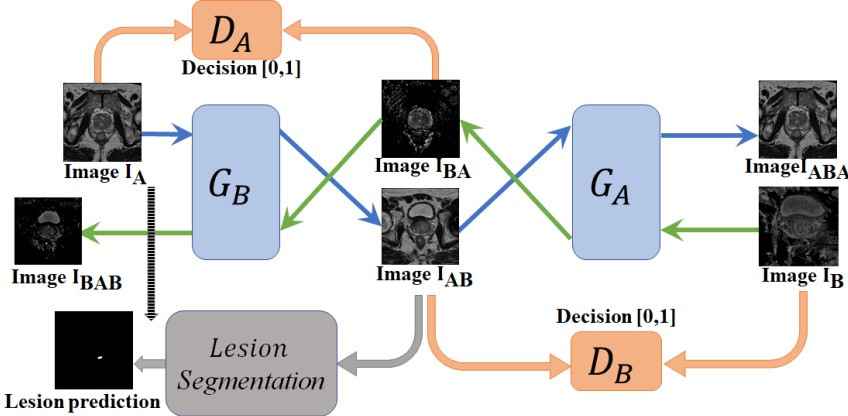

Figure 1: Our proposed framework to generate Hybrid ADC.

## 4. Results and Concluding Remarks

We have used the anonymized T2w-MR and ADC maps of 354 patients for training our algorithm and tested our proposed solution on 74 patients. According to the expert radiologist we could satisfy the requirements for producing HADC with potential for substitution in cases without using endocortical coil for imaging. A sample visual representation of our obtained results are demonstrated in Fig. 2, with complete anatomical harmony of lesion contrasting HADC and T2w-MRI.

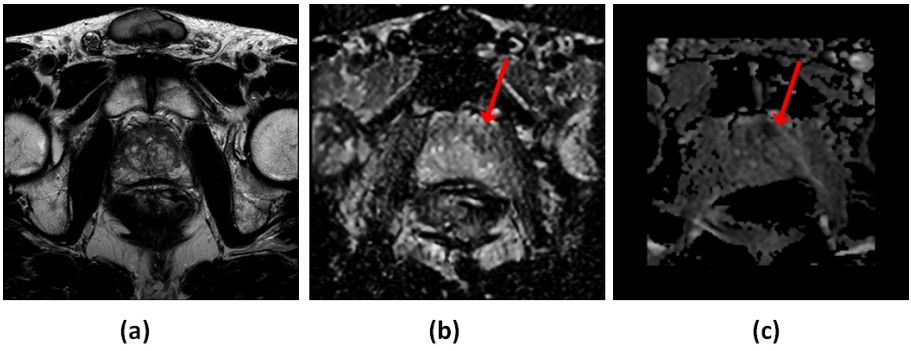

(a)                    (b)                    (c)

Figure 2: (a) T2w-MR, (b) our generated HADC map, and (c) acquired ADC map.

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

## Appendix A. Loss functions

$$
\begin{aligned}
\mathrm{G_{loss}} =& \mathrm{MSE}(\mathbf{1}, D_B(I_{AB})) + \mathrm{MSE}(\mathbf{1}, D_A(I_{BA})) + \mathrm{BCE}(I_{ABpred}, I_{Amask}) \\
& +.1[\mathrm{MSE}(I_{AB}, I_B) + \mathrm{MSE}(I_{AB}, I_A)] + 10[\mathrm{MSE}(I_{ABA}, I_A) + \mathrm{MSE}(I_{BAB}, I_B)].
\end{aligned} \tag{1}
$$

$$
\mathrm{D_{loss}} = \mathrm{MSE}(\mathbf{1}, D_B(I_B)) + \mathrm{MSE}(\mathbf{0}, D_B(I_{AB})) + \mathrm{MSE}(\mathbf{1}, D_A(I_A)) + \mathrm{MSE}(\mathbf{0}, D_A(I_{BA})). \tag{2}
$$

In these definitions (Eq. 1 and 2), MSE(.) and BCE(.) respectively stand for element-wise Mean Square Error and Binary Cross Entropy of two input arguments. Accordingly, $\mathbf{1}$ and $\mathbf{0}$ imply vectors of similar shape as the second argument to MSE() function. Also, $D_B(I_B)$ is the resulted feature vector of the discriminator.

