# OpenReview forum: "Hybrid Apparent Diffusion Coefficient (HADC) Map"
_MIDL.io/2020/Conference — Submitted to MIDL 2020_

### Official Review · AnonReviewer1 · 2020-02-23
**Application of cycleGAN image to image translation for ADC map generation from T2w MRI**

**Rating:** 1
**Confidence:** 5

**Review:**

Pros: The paper presents an excellent clinical motivation for the work and introduces the problem really well.

Cons:
However, there isn't anything methodologically new here. Its just an application of cycleGAN to this problem with the addition of Boundary loss (BCE loss) presented in the appendix.

Application of a well-known technique to this problem is not completely unreasonable if there are sufficient results. The experimental result is only one qualitative example. Its unclear how good this qualitative result is. For instance, would a radiologist use the generated ADC in place of the acquired ADC map for MRI interpretation? This is fundamentally important and crucial for this approach.

Even if it wasnt possible to have radiologists assessment, it would be interesting to see how the generated ADC improves over acquired ADC in prostate cancer classification. This is not presented either.

---

### Official Review · AnonReviewer3 · 2020-03-10
**Generation of a hybrid structural ADC map using GANs**

**Rating:** 2
**Confidence:** 3

**Review:**

- Good summary of clinical problem in prostate cancer and need for "hybrid" ADC map with more structural information
- Use of GANs to "translate" between ADC and T2 maps, but the exact logic underlying what the GAN is optimizing for is not well explained.
- Impact of post-processing of T2w MRI is unclear.
- No quantitative evaluation, hard to tell how effective the approach is.

---

### Official Review · AnonReviewer2 · 2020-03-15
**Generate ADC from T2 prostate MRI**

**Rating:** 1
**Confidence:** 5

**Review:**

The authors claim to have developed a method to generate an ADC image from T2Weighted prostate MRI.
The assumption is that T2-weighted imaging contains information to generate ADC images. I find this assumption absurd. Prostate MRI is well researched and the information in t2 and DWI imaging is clearly distinct and both are required. Depending on the zone either T2 or DWI is required to diagnose prostate cancer. (Read PIRADS). Why not claim this for all MRI imaging? Just acquire one sequence and you'll generate all other images of a patient!

---

### Meta-Review · Area_Chair1 · 2020-03-27
**MetaReview of Paper98 by AreaChair1**

**Rating:** 1

**Metareview:**

I agree with the reviewers that the assumption that T2-weighted images contain information to generate ADC images is absurd. Synthesized images can reflect standard ADC values per tissue but not patient-specific values.


Besides, Authors mention "using our proposed LA-CGAN (Zhu et al., 2017)", so not an anonymous submission.

lack of quantitative results.

**Paper Type:**

methodological development

---

### Decision · Program_Chairs · 2020-04-11

Reject